# Correlations among Quality Characteristics of Green Asparagus Affected by the Application Methods of Elevated CO₂ Combined with MA Packaging

**Li-Xia Wang** [1,*], **In-Lee Choi** [1,2] **and Ho-Min Kang** [1,3,*]

[1]   Division of Horticulture and Systems Engineering, Kangwon National University, Chuncheon 24341, Korea; cil1012@kangwon.ac.kr

[2]   Agricultural and Life Science Research Institute, Kangwon National University, Chuncheon 24341, Korea

[3]   Interdisciplinary Program in Smart Agriculture, Kangwon National University, Chuncheon 24341, Korea

*   Correspondence: 2018lisa@kangwon.ac.kr (L.-X.W.); hominkang@kangwon.ac.kr (H.-M.K.);
    Tel.: +82-33-250-6425 (H.-M.K.); Fax: +82-33-259-5562 (H.-M.K.)

**Abstract:** This research investigated the effects of continuous elevated CO₂ (20%, (*v/v*)) application or a 3 day CO₂ pretreatment followed by modified atmosphere (MA) or micro-perforated (MP) packaging on the postharvest quality of asparagus. The combination of CO₂ pretreatment with MA packaging (Pre-MA) inhibited the yellowing of asparagus and fresh weight loss (FWL), whereas stem firmness slightly increased with all elevated CO₂ treatments. CO₂ pretreatments increased antioxidant activity in the stem, but not in the tip, in contrast to the continuous flow CO₂ (Flow-CO₂) treatment. The phenolic and flavonoid contents increased in the elevated CO₂ pretreatments and Flow-CO₂ treatment. The elevated CO₂ treatments, especially Flow-CO₂, inhibited the development of microorganisms, and the treated asparagus did not decay. Pre-MA and Flow-CO₂ treatments were more effective in maintaining the visual quality and retarding the off-odor of asparagus. Furthermore, significant correlations between sensory quality characteristics and physiological-biochemical attributes were recognized; three principal components were extracted and they explained 86.4% of asparagus characteristics. The results confirmed the importance of visual quality, off-odor, firmness, color parameters, SSC and total phenolic content. In conclusion, elevated CO₂ pretreatment followed by MA packaging (Pre-MA) was beneficial for extending asparagus cold storage shelf life, and Flow-CO₂ was the best treatment for inhibiting postharvest decay.

**Keywords:** elevated CO₂; modified atmosphere package; sensory and physiological-biochemical characteristics; total phenol; DPPH

## 1. Introduction

In recent years, the consumption of green asparagus (*Asparagus officinalis* L.) has been increasing due to its good eating quality, special flavor, and abundance of nutritional value. Unfortunately, rapid quality deterioration, including toughening, off-odor development, shriveling, and fresh weight loss due to higher respiration [1] and metabolic activities occurs. The rapid postharvest loss of green asparagus quality poses a challenge to the development of effective methods to retard the decline of quality and prolong shelf life. The application of modified atmosphere packaging (MAP) at low temperature as an effective technology has been reported to be beneficial for extending the shelf life and maintaining the quality of vegetables and fruits by reducing respiration rate and fresh weight loss, delaying ripening and minimizing physiological disorders and decays [2,3]. In lotus (*Nelumbo nucifera*), MAP treatments delayed the browning of roots, which involves changes in phenols [4]. The ambient gas

levels in postharvest storage are important as they relate to respiration rate and physiological changes [5]. Prestorage or continuous elevated $CO_2$ treatments have been used for many fruits and vegetables to improve quality, inhibit the development of microbial groups and prolong shelf life. Short-term application of high $CO_2$ maintained the solids content and firmness of white asparagus spears, which is related to change of metabolic activity and respiration rate [6]. In grapes (*Vitis vinifera* L.), high $CO_2$ (20 kPa) pretreatment improved the appearance of bunches and maintained berry quality [7], reduced total decay and induced the accumulation of three small heat shock proteins (HSPs) [8]. Nutrients such as soluble sugar, soluble protein, and free amino acid content were also increased by elevated $CO_2$ in kidney beans (*Phaseolus vulgaris*) [9]. Treatment with 20% $CO_2$ prolonged the strawberry (*Fragaria X ananassa* Duch.) storage period to 12 days, reduced the energy charge related to the decline of NADH/NAD$^+$ and caused the accumulation of γ-aminobutyric acid (GABA) [10]. Fungal decay was prevented, and no fermentation was observed with 20% $CO_2$ pretreatment for two days of fresh goji (*Lycium* spp.) berries [11]. High $CO_2$ pretreatment has also been used for white asparagus, which confirmed that the effects of higher $CO_2$ (10%) on biochemical and textural properties were determined by the storage temperature [6]. In a previous study, we reported that vacuum packages supplemented with 60% (*v/v*) $CO_2$ induced a higher soluble solids content, lower firmness, greener color, and less weight loss, but more off-odor [12]. Considering the low risk and high benefit of high $CO_2$ treatment in many fruits and vegetables, the use of prestorage or continuous treatments with elevated $CO_2$ has been proposed as a suitable tool to control the development of microorganisms and maintain green asparagus quality. However, few studies have been carried out on the effect of elevated $CO_2$ treatment on green asparagus postharvest quality and changes in phytochemical component and antioxidant activity.

This work aimed to analyze the effectiveness of continuous 20% $CO_2$ treatment and $CO_2$ pretreatment followed by MAP in controlling the development of microbial groups and its effect on quality attributes, antioxidant ability, and physiological-biochemical characteristics during green asparagus storage period at 4 °C.

## 2. Materials and Methods

### 2.1. Plant Materials, CO$_2$ Treatments, Packaging Material and Storage Conditions

Green asparagus (*Asparagus officinalis* L. cv. 'Welcome') was cultivated and provided by a local farm (Yanggu-gun, Gangwon-do, Korea, lat. 38°12′33.00″ N, long. 127°13′3.00″ E) in May 2019. Green asparagus spears were harvested and transported to the Postharvest Physiology and Distribution Laboratory, Kangwon National University, on the same day and stored in a 4 °C refrigerator. Healthy and uniform (20–30 g/spear, 1.4 ± 0.1 cm diameter and 24.0 ± 1.0 cm length) green asparagus spears were selected and randomized for elevated $CO_2$ treatments in this study.

A passive modified atmosphere (MA) package with a 10,000 cc/m$^2$·day$^{-1}$·atm$^{-1}$ oxygen transmission rate (OTR) (Dae Ryung Precision Packaging Industry Co., Ltd., Gwangju-si, Korea) and a micro-perforated (MP, 34 holes on 10,000 cc/m$^2$·day$^{-1}$·atm$^{-1}$ MA films perforated by drilling with 0.6 mm diameter tips) package were selected to package asparagus spears. A 20% (*v/v*) carbon dioxide ($CO_2$, released from dry ice) treatment was applied in a sealed plastic box (30 × 21 × 15 cm) An air pump (DK-3000, Dae-Kwang Electronics Co., Ltd, Korea) was used to control the $CO_2$ concentration and $CO_2$ concentration was measured every 3 hours.

Initially, selected asparagus spears were randomly divided into two groups: pretreated with 20% $CO_2$ for 3 days and 20% $CO_2$ continuous application until the last day of cold storage (20 days). The 20% $CO_2$ pretreatment group was further divided into two groups: packaged with MA (Pre-MA) and MP (Pre-MP) packages. Spears treated with continuous 20% $CO_2$ (Flow-$CO_2$) were stored in a sealed plastic box (30–21–15 cm). Untreated groups packaged with MA (Cont-MA) and MP (Cont-MP) packages were used as the controls. All of the groups were stored at 4 ± 0.5 °C and 85 ± 5% relative

humidity (RH) for 20 days. Five replicates of 20 green asparagus spears each were used for each packaging group.

*2.2. Microbiology Analysis*

Microbiological analysis was performed on initial samples (Initial day), after 3 days of pretreatment (Treated day 3), and after 20 days of storage according to Wang [12] with some modifications. Fresh asparagus (2.0 g) was mixed with 18 mL sterilized distilled water using a stomacher blender (Powermixer, B&F Korea, Gimpo-si, Korea) set at the highest speed (level 10, 200 rpm) for 3 min. Then, the mixture was diluted by a factor of 1000. Next, 1.0 mL of the dilution was dropped on a microbiology Petri film plate (3 M Co., St Paul, MN, USA). Aerobic bacteria, yeast and mold, and *Escherichia coli* were cultivated for 72 h at 35 °C, 72 h at 25 °C, and 24 h at 35 °C, respectively. The development of total aerobic bacteria (TAB), yeast and mold (Y&M), and *E. coli* were measured using Petrifilm Plate Reader (3M Co., St. Paul, MN, USA). The number of microorganisms was represented by the base 10-logarithm of the colony-forming unit concentration (log $CFU \cdot g^{-1}$).

*2.3. Changes in Quality Parameters of Green Asparagus*

Firmness (N) was measured at two locations on each asparagus spear, at 5 cm from the tip and 8 cm from the base using a rheometer (Compac-100, Sun Scientific Co. Led.,Tokyo, Japan) with a probe (Ø 3.0 mm) at 1.0 mm/sec speed.

Green asparagus color variables were measured using a color-difference meter (Model CR-400, Konica Minolta Sensing, Inc., Japan) at 5 cm from the tip and at 8 cm from the stem for lightness (*L\**), Chroma (*C\**), redness *a\**, blueness *b\** and hue angle (h°). The total color difference was represented as *ΔE\**, and calculated as:

$$\Delta E* = \sqrt{(L*-L)^2 + (a*-a)^2 + (b*-b)^2} \tag{1}$$

where *L*, *a* and *b* were the color parameters of fresh asparagus spears without treatment (Initial day).

The total chlorophyll content was measured following Yoon [13] with slight modifications. Frozen asparagus samples (1.0 g) were chopped and mixed in 10 mL methanol and then incubated at 4 °C for 48 h to extract chlorophyll. The absorbances at 642.5 nm ($A_{642.5}$) and 660 nm ($A_{660}$) were measured using a UV–VIS spectrophotometer (UV mini model 1240, Shimadzu, Japan). The total chlorophyll content was calculated using the following formula:

$$\text{Total chlorophyll} \left(mg \cdot mL^{-1}\right) = 7.12 \times A_{660} + 16.8 \times A_{642.5} \tag{2}$$

Soluble solids content (SSC) was measured by a pocket refractometer (PAL-1, Atago, Tokyo, Japan). Asparagus samples were chopped up and extruded with gauze wrapping. The asparagus solution was directly dripped onto a pocket refractometer and the SSC result was indicated as °Brix at ambient temperature [13].

Fresh weight loss was measured according to the following formula:

$$\text{Weight loss}(\%) = \frac{\text{Initial fresh weight} - \text{Final fresh weight}}{\text{Initial fresh weight}} \times 100\% \tag{3}$$

Fresh asparagus was weighed and put in an oven (OF-21E, Jeio Tech Co., Ltd., Daejeon, Korea) for 10 min at 103 °C, dried to constant weight at 72 °C, and re-weighed. The water content of asparagus spears on the final day of storage at 4 °C was determined as follows:

$$\text{Water content}(\%) = \frac{\text{Fresh weight} - \text{Dry weight}}{\text{Fresh weight}} \times 100\%. \tag{4}$$

Sensory qualities, including visual quality and off-odor of green asparagus, were assessed by five skilled members from the "Postharvest Physiology and Distribution Laboratory" [13]. The visual

quality of the asparagus was assessed throughout the entire storage period and scored on a scale of 1 to 5 (1 = worst: yellowing, decay, shrinking, woodiness, bract opening, 2 = bad, 3 = good, 4 = better, 5 = best: an at-harvest appearance, no decay or defects, dark green, no shrinking or bract opening). Off-odor was assessed after 20 days of storage and scored on a scale of 0 to 5 (0 = no off-odor and 5 = strong off-odor). Asparagus with a visual quality score equal to or greater than 3 and an off-odor score equal to or less than 3.0 was determined to be marketable.

### 2.4. The Effect of High $CO_2$ on Phenol, Flavonoids and DPPH of Green Asparagus

Asparagus (2.0 g) was homogenized in 1% HCl-methanol (*v/v*) solution, the homogenate was placed in a 25 mL tube with distilled water and incubated for 20 min in the dark, and then it was filtered with 110 mm filter paper. The absorbance of the solution was measured at 280 nm and 325 nm [14]. The contents of total phenol and flavonoids were calculated as follows:

$$\text{Total phenol content} = \frac{\text{OD}_{280}}{\text{g}} \tag{5}$$

$$\text{Flavonoids content} = \frac{\text{OD}_{325}}{\text{g}} \tag{6}$$

Antioxidant activity was measured by the DPPH ($\alpha,\alpha$-diphenyl-$\beta$-picrylhydrazyl) method [15]. Green asparagus samples (2.0 g) were ground at 4 °C in 20 mL of methanol and then filtered with 110 mm filter paper, 1.0 mL of the filtered solution was mixed with 1.0 mL of a 0.4 mM DPPH in methanol solution, and the mixture was incubated in darkness for 30 min before measuring the absorbance at 516 nm.

$$\text{DPPH radical scavenging activity (\%)} = \left[1 - \frac{\text{sample A}_{516}}{\text{blank A}_{516}}\right] \times 100\% \tag{7}$$

### 2.5. Gas Conditions and Respiration Rate

By inserting a needle into the packages through a septum, 1.0 mL gas samples from the headspace of packages were collected. Carbon dioxide ($CO_2$) concentration in different treatments was measured with an infrared $CO_2/O_2$ analyzer (Model Check Mate 9900, PBI-Dansensor, Ringsted, Denmark). Ethylene ($C_2H_4$) concentration was measured with a GC-2010 Shimadzu gas chromatograph (GC-2010, Shimadzu Corporation, Japan) [12], equipped with BP 20 Wax column (30 m × 0.25 mm × 0.25 um, SGE analytical science, Australia) and a flame ionization detector (FID). The detector and injector operated at 127 °C, the oven was at 50 °C, and the carrier gas ($N_2$) flow rate was 0.67 mL·$s^{-1}$. $CO_2$ and $C_2H_4$ concentration changes in MA and MP packages were determined on the first and third days of cold storage and subsequently were measured every 5 days up to 20 days of storage at 4 °C. The change in $CO_2$ concentration of pretreated, continuous and control groups was measured every 1 hour at ambient temperature and determined in five duplicate packages. The respiration rate was expressed as mL $CO_2$ $kg^{-1}$ $h^{-1}$.

### 2.6. Statistical Data Analysis

The software Microsoft Excel 2019, R program (Version 4.0.2, 2020-06) and IBM SPSS Statistics (24, IBM Corp., Armonk, NY, USA) was used to statistically analyze the data. The effects of elevated $CO_2$, MA/MP packaging, and $CO_2$ × package interaction were analyzed using two-way ANOVA. All results are expressed as the means (n = 5) and their standard error (SE). Significant differences were tested with ANOVA (one-way analysis of variance) and Duncan's Multiple Range Test at $\alpha < 0.05$. All experiments were performed with at least three independent repetitions and repeated three times. Principal component analysis (PCA) scores and correlation coefficients between sensory quality and physiological and biochemical performance were analyzed using Rstudio version 4.0.2 (Team 2020), with distance measurements by Pearson's correlation coefficient test.

## 3. Results

*3.1. Changes in Quality Parameters of Green Asparagus*

3.1.1. Fresh Weight Loss (FWL)

Fresh weight loss developed in all treatments, with the increase greater in MP packages than in MA packages during the entire storage period (Figure 1). Fresh weight loss of green asparagus was $2.10 \pm 0.06\%$ in Cont-MP and $1.55 \pm 0.09\%$ in Pre-MP, whereas in MA it was $0.58 \pm 0.05\%$ in Cont-MA and $0.48 \pm 0.02\%$ in Pre-MA, all of which significantly differed. Flow-$CO_2$ treatment showed the highest FWL of $2.30 \pm 0.05\%$, which was probably due to the lack of a film package during cold storage. It is worth noting that although the weight loss of different treatments was significantly different, all treatments had weight loss below 2.5% and were very similar to each other. FWL was well below the limit of marketable acceptance of 8%. In addition, elevated $CO_2$ inhibited FWL with a lower loss than that of the control. Significant differences were found between Cont-MP and Pre-MP. Pre-MA had the lowest FWL after 20 days of storage at 4 °C. The inhibition of FWL by high $CO_2$ pretreatments was also observed with goji berries, with significantly lower weight loss than that of control fruit [11]. Previous studies have reported that water content is important for modifying the quality and storage period of fruits and vegetables [16,17].

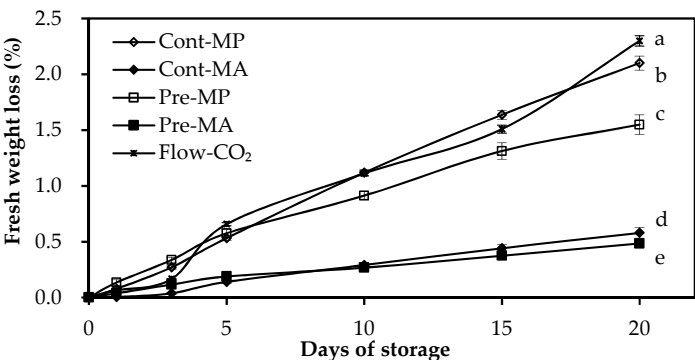

**Figure 1.** Fresh weight loss (%) during green asparagus cold storage as influenced by elevated $CO_2$ treatments. Values are means of five repetitions ± standard errors (SE). Different letters over bars indicate significant differences between treatments by Duncan's Multiple Range Test ($\alpha < 0.05$).

3.1.2. Firmness

Compared to the initial samples, asparagus firmness increased after 20 days of storage (Figure 2A). Considering the controls and all $CO_2$ treatments, a notable increase in firmness was observed in the stem with the initial value of 12.3 N. Compared to the control, Pre-MP and Flow-$CO_2$ treatments significantly increased the firmness on the last storage day and showed the highest firmness stem (~19.3 N). A significant difference in asparagus stem firmness was observed with elevated $CO_2$ pretreatment. The increase in firmness was less in MA than MP packages. This suggested that Pre-MA was more efficient at inhibiting the development of firmness during postharvest storage. This was in keeping with the results that the asparagus spear water content was lower in the Flow-$CO_2$ than the control and $CO_2$ pretreatments (data not shown). A similar negative relationship between firmness and water content in carrots has been reported [18]. The increase in firmness in asparagus was correlated with an increase in crude fibers, the development of lignin and the activity of phenylalanine ammonia-lyase [19]. Likewise, our results revealed significant differences in firmness only in the stem between the elevated $CO_2$ treatments and the control. This result suggests that $CO_2$ treatment influenced the wound response from cutting damage. Verlinden [20] reported a higher respiration rate, higher moisture loss and enhanced $CO_2$ diffusion during the wound response in the asparagus stem, which may lead to higher firmness.

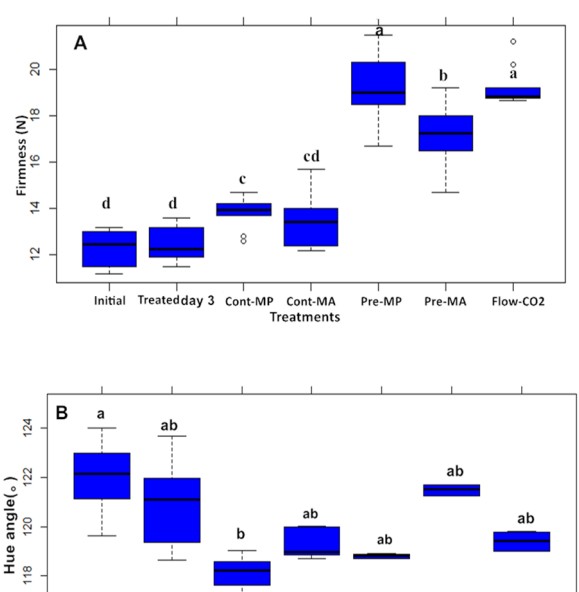

**Figure 2.** Effects of elevated $CO_2$ treatments on (**A**) firmness and (**B**) hue angle in asparagus stem on the initial day, treated day 3 and after 20 days of storage at 4 °C. The different letters represent significant differences by Duncan's Multiple Range Test ($\alpha < 0.05$ level). Box plots consist of the median, the lowest and the highest values, the outliers, and the approximate quartiles of our parameters.

### 3.1.3. Color Parameters

Color is one of the most important factors for postharvest biology and sensory evaluation of green asparagus. Yellowing (i.e., lower hue angle) of green asparagus indicates degradation of quality during storage [21]. Yellowing was obvious and showed a trend between the initial day and after 20 days of storage (Figure 2B) with only Cont-MP lower than the Initial value. The deviation from the raw material total color was represented as $\Delta E^*$ and the tip was significantly less than the other treatments at 20 days in the Flow-$CO_2$ treatment, and it did not differ from $CO_2$-treated day 3 (Table 1). For stems, the lowest color difference was observed for the Pre-MP treatment at 20 days, with a value of 5.19, which was less than the treated day 3 values.

**Table 1.** The means of chlorophyll content and total color deviation ($\Delta E^*$) obtained from tip and stem of green asparagus in initial, treated day 3 and on the 20th day of storage at 4 °C.

| Treatments | Chlorophyll Content (mg·mL$^{-1}$) | | $\Delta E^*$ | |
| --- | --- | --- | --- | --- |
| | **Tip** | **Stem** | **Tip** | **Stem** |
| **Initial day** | 5.66 a | 2.18 a | - | - |
| **Treated day 3** | 4.84 b | 1.98 c | 6.63 c | 6.54 b |
| **Cont -MP** | 3.23 f | 1.12 g | 9.35 b | 10.23 a |
| **Cont -MA** | 3.09 g | 1.74 f | 8.34 bc | 10.30 a |
| **Pre-MP** | 3.76 d | 1.84 e | 13.38 a | 5.19 c |
| **Pre-MA** | 4.68 c | 2.11 b | 7.29 bc | 8.91 a |
| **Flow-$CO_2$** | 3.62 e | 1.89 d | 6.25 c | 8.20 ab |

Values are means of five repetitions (n = 5). Different letters within a column indicate significant differences between treatments by Duncan's Multiple Range Test ($\alpha < 0.05$)

The green color is mainly determined by chlorophyll content [22]. The chlorophyll content was consistent with the hue angle value in all treatments (Table 1). Initial day samples had the highest chlorophyll content and subsequently exhibited a progressive decrease to the last day of cold storage. Interestingly, lower chlorophyll content was observed in the controls at the start and after cold storage. There was a significant difference among all the treatments; the Pre-MA treatment showed the highest chlorophyll content, and the Cont-MP treatment had the lowest content. The breakdown of chlorophyll is closely related to the remobilization of chloroplast proteins, lipids and metals during senescence [23]. An inhibitory effect of elevated $CO_2$ on the decrease in chlorophyll content of broccoli florets after harvest has been reported [24]. High $CO_2$ treatment affected anthocyanin stability and color expression in strawberries by changing the pH, while MA packaging with 15% $CO_2$ treatment maintained the anthocyanin stability of strawberries [25]. In the current study, Pre-MA treatment evidenced greater inhibition of yellowing during green asparagus cold storage. The mechanism of elevated $CO_2$ and MA package effects may have been partially based on a decrease in chlorophyllase activity that delays the degradation of chlorophyll because of the decrease in metabolic activity. This is consistent with the results that high $CO_2$ delayed chlorophyll degradation and anthocyanin accumulation by inhibiting chlorophyllase activity and downregulating the expression of *FaChl b reductase*, *FaPAO* and *FaRCCR*, which are related to chlorophyll state in postharvest storage of strawberry fruits [26]. Likewise, cold shock delayed the degradation of chlorophyll by suppressing the activity of chlorophyllase in cucumber [27].

### 3.1.4. Soluble Solids Content (SSC)

Total soluble solids content (SSC) are mainly sugars, which tended to decrease towards the end of shelf life in all of the samples [28]. A low soluble solids content was observed in all treatments and significantly differed among treatments (Table 2). On treated day 3, SSC had decreased to 5.58 °Brix compared to the initial day samples at 6.19 °Brix. The decrease was probably related to respiration, which is the key factor in converting soluble solids into energy [29]. There was no significant difference between the control (5.30 °Brix), Flow-$CO_2$ (5.23 °Brix) and Pre-MA (5.03 °Brix) treatments after 20 days of storage. However, the lowest SSC was in the Pre-MP treatment, with a content of 4.37 °Brix on the last day of cold storage. Flow-$CO_2$ treatment without film packages slowed the decrease in the SSC. Li and Zhang [30] reported that the water loss of green asparagus also contributed to an increasing soluble solids content, which is consistent with the water content of asparagus in this study (data not shown). A similar result was reported with grapes, in which high $CO_2$ treatment inhibited the decrease in the SSC due to an inhibition of normal postharvest metabolic activity [31].

**Table 2.** Effects of elevated $CO_2$ treatments on green asparagus soluble solids content (SSC), total phenolic and flavonoid contents, and DPPH-radical scavenging activity, on the initial day, treated day 3 and after 20 days of storage at 4 °C.

| Treatments | SSC (°Brix) | Total Phenol (U·g$^{-1}$) | Flavonoids (U·g$^{-1}$) | DPPH (%) |
|---|---|---|---|---|
| **Initial day** | 6.19 a | 0.72 c | 0.68 e | 85.67 c |
| **Treated day 3** | 5.58 b | 0.82 a | 0.91 a | 92.67 a |
| **Cont-MP** | 4.93 c | 0.67 d | 0.58 g | 56.97 e |
| **Cont-MA** | 5.30 bc | 0.79 b | 0.61 f | 73.73 d |
| **Pre-MP** | 4.37 d | 0.80 ab | 0.72 d | 71.88 d |
| **Pre-MA** | 5.03 c | 0.81 a | 0.74 c | 88.64 b |
| **Flow-$CO_2$** | 5.07 c | 0.81 a | 0.76 b | 91.18 ab |

Values are means of five repetitions (n = 5) and each letter in the same column indicate the significant differences by Duncan's Multiple Range Test ($\alpha < 0.05$).

### 3.2. The Effect of High $CO_2$ on Phenol, Flavonoids and DPPH of Green Asparagus

Phenolic compounds such as flavonoids and anthocyanin are important naturally occurring antioxidant compounds from many plants. The phenolic content could be influenced by biotic and

abiotic factors [32,33]. Moreover, the antioxidant capacity of phenolic compounds is related to the ability to scavenge DPPH radicals in an assay [34]. In this study, total phenol and flavonoids content and DPPH radical scavenging activity were measured on the initial day, treated day 3 and after 20 days of storage (Table 2). The total phenolic and flavonoid content in the initial day samples were 0.72 $U·g^{-1}$ and 0.68 $U·g^{-1}$, respectively. Total phenolic and flavonoid content increased in the $CO_2$ pretreatments on treated day 3, which showed the highest absorbance values of 0.82 $U·g^{-1}$ and 0.91 $U·g^{-1}$, respectively. At 20 days, the levels of total phenolics under elevated $CO_2$ pretreatments had decreased compared to treated day 3 but increased compared to the initial day. The lowest total phenolic and flavonoid contents were observed in the Cont-MP treatment, with values of 0.67 $U·g^{-1}$ and 0.58 $U·g^{-1}$, respectively. The difference between the control groups and all elevated $CO_2$ treatments was pronounced, although no difference was found between elevated $CO_2$ pretreatments and Flow-$CO_2$ treatment in total phenolics but not in flavonoids. Although elevated $CO_2$ treatments slowed the loss of phenolic components, a decrease occurred after longer storage in all elevated $CO_2$ treatments and control groups which is similar to changes in strawberry [35].

Postharvest $CO_2$ enrichment stimulated antioxidant activities and phenolic content in lettuce [36]. The beneficial effect may be because elevated $CO_2$ treatment is a kind of abiotic stress promoting the synthesis and accumulation of phenolic as a physiological response. Previously, $CO_2$ storage had marked effects on phenolic metabolites and quality, while modified atmospheres (MA) had a positive effect on phenolic-related quality [37]. High $CO_2$ may allow for the removal of free radicals, which are associated with an increase of antioxidant capacity [38]. The DPPH radical scavenging activity of green asparagus showed a similar change in pattern as the phenolic and flavonoid contents in response to the different treatments (Table 2). The highest DPPH value of 92.67 was a significant increase after $CO_2$ pretreatment. Flow-$CO_2$ treatment sustained the antioxidant activity through 20 days, with a DPPH value of 91.18. Based on these points, the Flow-$CO_2$ treatment was noted for improving asparagus qualities and maintaining the higher bioactive compound levels for long-term postharvest storage.

### 3.3. Gas Conditions in Packages and Respiration Rate

MA and MP packages involve the process of changing the atmospheric gas composition surrounding the green asparagus, especially carbon dioxide and oxygen concentration, by controlling gas exchange between the inside of the packages and the ambient air. The gas levels exhibited different concentration changes between MA packages and MP packages caused by their difference in oxygen transmission rates (OTRs). As shown in Figure 3A, the $CO_2$ concentration increased rapidly on the first day of cold storage in all packages and subsequently remained stable until a small decrease at 15 days of storage. MA packages showed higher $CO_2$ concentrations than MP packages throughout the whole storage period, 4–6% versus 0–2%, respectively. A $CO_2$ level above 10% and $O_2$ level below 3% can produce injuries to green asparagus at the optimal temperature [39,40]. Compared to the gas composition in MP packages, a better gas composition was observed in MA packages. The recommended gas concentration for green asparagus storage at 5 °C is 5–10% $O_2$, similar to ambient composition [41], and a $CO_2$ level of 4–6% probably decreases the respiration rate and inhibits anaerobic metabolism. The concentration of ethylene in all packages first increased (Figure 3B), and was followed by a decrease, finally becoming stable. It is possible that this was because elevated $CO_2$ reduced the respiration rate and inhibited the biosynthesis and negative effects of ethylene.

The respiration rate measured on the initial day was 26.26 mL $kg^{-1}$ $h^{-1}$ $CO_2$, while elevated $CO_2$ pretreatment showed a value of 24.62 mL $kg^{-1}h^{-1}$ $CO_2$ (Figure 3C). Comparing the respiration rate of the Flow-$CO_2$ treatment on the final day (35.67 mL $kg^{-1}$ $h^{-1}$ $CO_2$) to the initial value, the significant difference could be ascribed to the long period of storage and the influence of elevated $CO_2$ since the respiration rate increased. Asparagus is classified as perishable due to a high respiration rate of 60 mg $kg^{-1}$ $h^{-1}$ $CO_2$ (at 5 °C) and rapid degradation of quality [5,42].

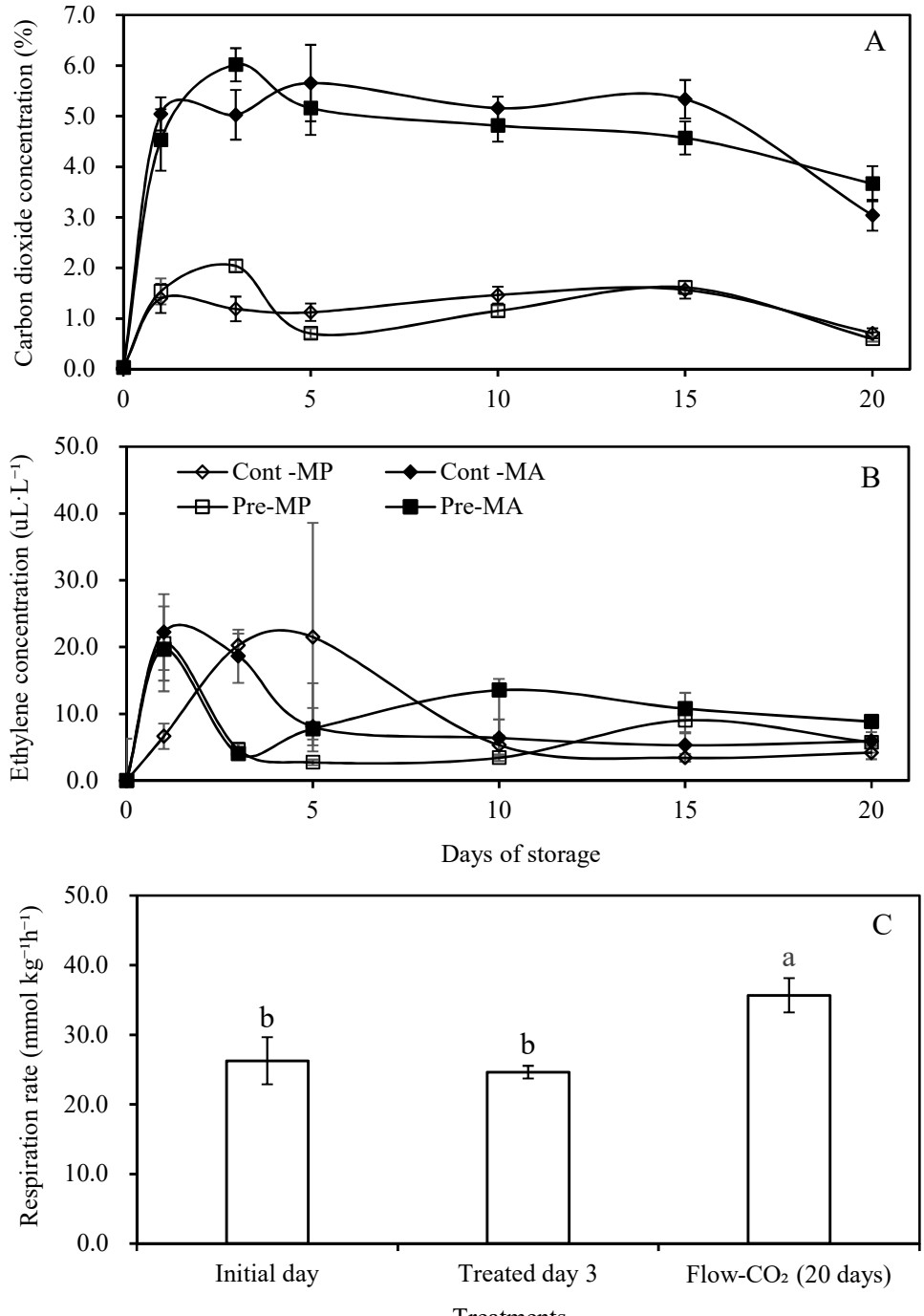

**Figure 3.** Changes in carbon dioxide (**A**) and ethylene (**B**) concentrations in modified atmosphere (MA) and micro-perforated (MP) packages during the postharvest storage, and the respiration rate (**C**) of green asparagus as influenced by high $CO_2$ pre and continuous treatments. Values are means of five repetitions ± standard errors (SE). Different letters over bars indicate significant differences among all treatments by Duncan's Multiple Range Test ($\alpha < 0.05$).

*3.4. Microorganism Analysis*

Compared to the initial value of 3.26 log CFU·$g^{-1}$, the growth of total aerobic bacteria (TAB) was inhibited by elevated $CO_2$ pretreatment with 2.10 log CFU·$g^{-1}$ (Table 3). Bacterial growth is critical in the spoilage of green asparagus, as significant differences in aerobic bacterial count between the initial and final storage days occurred. After 20 days of storage, TABs significantly increased,

especially in Cont-MP, which had the highest count of 5.76 log $CFU \cdot g^{-1}$. Likewise, regardless of the packages, there was no significant difference in TABs between the $CO_2$ pretreatments and control groups. However, Flow-$CO_2$ with a TAB count of 2.16 log $CFU \cdot g^{-1}$ was not significantly different from the initial or treated day 3 samples, which indicated that the Flow-$CO_2$ treatment controlled the growth of these microorganisms and maintained lower counts up to the 20th day at 4 °C. The initial *E. coli* counts of green asparagus were 2.71 log $CFU \cdot g^{-1}$, and the counts increased through 20 days of storage in all treatments. Cont-MP and Cont-MA reached 4.31 and 4.20 log $CFU \cdot g^{-1}$, respectively. Treated day 3 did not show a sterilization effect on *E. coli* with 3.50 log $CFU \cdot g^{-1}$. However, a significant effect of elevated $CO_2$ pretreatments on *E. coli* was observed with lower counts after 20 days of storage at 4 °C than at treated day 3 (Table 3). Likewise, 3.01 log $CFU \cdot g^{-1}$ was observed under the Flow-$CO_2$ treatment on the last day of storage (20 days). The initial yeast and mold (Y&M) count before and after elevated $CO_2$ pretreatment was 0.00 log $CFU \cdot g^{-1}$ (Table 3). In contrast, Y&M increased significantly in the Cont-MP treatment on the last day of cold storage, with 2.10 log $CFU \cdot g^{-1}$. Notably, all elevated $CO_2$ treatments showed an encouraging lack of Y&M with counts of 0.00 log $CFU \cdot g^{-1}$. This was consistent with the result that high $CO_2$ was beneficial in controlling microbial growth in fresh-cut mango (*Mangifera indica*) cubes [43]. According to a previous study, short-term exposure to high levels of $CO_2$ modified the pathogenesis-related protein defense response and the low molecular mass chitinase, which renders grapes less susceptible to fungal infection [44]. It has also shown that elevated $CO_2$ retards the growth of microorganisms, which is lethal to spores and damages the cell wall of the microorganisms [45]. Likewise, the development of microorganisms inhibited by high $CO_2$ is probably due to its dissolution in the aqueous phase of food products, which causes intracellular acidification, inhibition of enzymatically catalyzed reactions and enzyme synthesis, and interaction with the cell membrane [46].

**Table 3.** Effects of elevated $CO_2$ pretreatments in MA and MP packages and continuous $CO_2$ treatments on the growth of microorganisms in green asparagus stored at 4 °C up to 20 days.

| Treatments | Growth of Microorganisms ( log $CFU \cdot g^{-1}$) | | |
|:---:|:---:|:---:|:---:|
| | Total Aerobic Bacteria | *E. coli* | Yeast and Mold |
| Initial day | 3.26 bc | 2.71 c | 0.00 b |
| Treated day 3 | 2.10 c | 3.50 b | 0.00 b |
| Cont-MP | 5.76 a | 4.31 a | 2.10 a |
| Cont-MA | 4.58 ab | 4.20 a | 0.00 b |
| Pre-MP | 4.75 ab | 3.34 b | 0.00 b |
| Pre-MA | 4.56 ab | 3.45 b | 0.00 b |
| Flow-$CO_2$ | 2.16 c | 3.01 bc | 0.00 b |

Values are means of five repetitions (n = 5) and different letters in the same column indicate significant differences by Duncan's Multiple Range Test ($\alpha < 0.05$) between means.

### 3.5. Sensory Quality and Relative Impacts of Asparagus Quality Attributes

On the last storage day, asparagus quality characteristics were assessed. Figure 4 illustrates the relationships among important quality traits. Visual quality, off-odor, fresh weight loss, and stem firmness (F-Stem) were affected significantly by elevated $CO_2$ and different packages. In contrast, tip firmness (F-Tip), hue angle in the tip (h°-Tip) and stem (h°-Stem), soluble solids content (SSC), and water content were not affected by elevated $CO_2$ and different packaging treatments. Visual quality and off-odor are major factors that determine consumption. *p*-values in this study demonstrated that elevated $CO_2$ treatments and different packages had significant effects on nearly every quality characteristic (Table 4). The elevated $CO_2$ main effect, package main effect and $CO_2 \times$ package interaction significantly affected fresh weight loss of green asparagus whereas no effect ($p \geq 0.05$) was observed for water content. However, the elevated $CO_2$ main effect and interaction of $CO_2 \times$ package effects did affect tip hue angle (h°-Tip), and tip firmness (F-Tip).

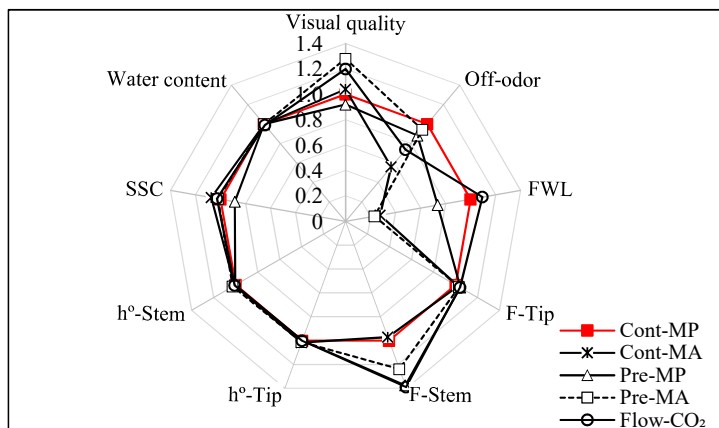

**Figure 4.** Spider graphs illustrating the relative impacts of elevated $CO_2$ treatments followed by MP or MA packaging on green asparagus quality characteristics after 20 days of storage at 4 °C. FWL: fresh weight loss; F-Tip: Tip firmness; F-Stem: Stem firmness; h°-Tip: Tip hue angle; h°- Stem: Stem hue angle; SSC: soluble solids content

**Table 4.** Analysis of variance results for the effects of elevated $CO_2$ × packaging (MP or MA) interaction green asparagus quality attributes after 20 days of storage at 4 °C.

| Quality Attributes | $CO_2$ | Package | $CO_2$ × Package |
|---|---|---|---|
| | *p*-Value | *p*-Value | *p*-Value |
| Visual quality | 0.06 | 0.03 [z]* | 0.07 |
| Off-odor | 0.04 * | 0.03 * | 0.00 ** |
| Fresh weight loss | 0.00 ** | 0.00 ** | 0.00 ** |
| Water content | 0.00 ** | 0.59 | 0.47 |
| SSC | 0.00 ** | 0.00 ** | 0.00 ** |
| F-Tip [y] | 0.32 | 0.77 | 0.41 |
| F-Stem | 0.00 ** | 0.03 * | 0.16 |
| h°-Tip | 0.25 | 0.02 * | 0.43 |
| h°-Stem | 0.00 ** | 0.00 ** | 0.00 ** |

[z]* = $p < 0.05$; ** = $p < 0.01$. [y] F-Tip: Tip firmness; F-Stem: Stem firmness; h°-Tip: Tip hue angle; h°- Stem: Stem hue angle.

Visual quality was estimated according to the extent of shrinkage, water loss, yellowing, bract opening, and fungal attack. In the present study, marketable visual quality was found in the Pre-MA and Flow-$CO_2$ treatments with scores of 3.2 and 3.0, respectively. Regarding the packages, the control groups lost retail value, and there was no significant difference between the control and Pre-MP treatments. The off-odor scores in the Cont-MA and Flow-$CO_2$ treatments were low and maintained commercial acceptability (off-odor score below 3.0), with scores of 1.9 and 2.5, respectively, after 20 days of storage. Regardless of packages, elevated $CO_2$ pretreatments lost marketable value with a strong off-odor. The accumulation of odor in green asparagus is caused by methanol and acetaldehyde, which are related to anaerobic respiration, microorganism decay and secondary metabolism [47,48]. Modified atmosphere (MA) packages reduced the production of off-odor by modifying the gas composition of packages and maintaining the adopted $CO_2$ (5–12%) content [5]. This is consistent with the growth of microorganisms in this study. Considering the better visual quality and slight off-odor, the notable efficiency of Flow-$CO_2$ treatment on asparagus quality and shelf life was confirmed in this study.

*3.6. Correlations and Principal Components Analysis*

The correlations between sensory quality characteristics and physiological and biochemical attributes after 20 days of storage at 4 °C are shown in Figure 5. A significant positive association between visual quality and soluble solids content (SSC), firmness (F), hue angle (h°), chlorophyll

content (Chl), and DPPH in the tip (DPPH-Tip). Negative correlations between visual quality and off-odor, water content (WC), total phenolic content in stem (TP-Stem), total flavonoid content (Fla), and DPPH in the stem (DPPH-Stem) occurred. Likewise, the off-odor showed negative correlations with SSC, h°, Chl, and Fla. This was consistent with the performance of green asparagus in different treatments, i.e., the greener the color, the higher the soluble solids content and chlorophyll, the better the visual quality value and the less the off-odor. Another interesting negative correlation between firmness and WC and SSC was observed. A change in firmness is related to the change of cell structure caused by the transformation of SSC and storage sugars in cell walls [49]. The changes in cell structure affect the transpiration of plants which relates to water content and fresh weight loss [27]. In addition, negative correlations were observed between FWL and h° and Chl. Another important finding was the association between sensory quality and physiological-biochemical characteristics between tips and stems of asparagus, which could be related to the at-harvest wounding of the stem (the relations between the DPPH and TP).

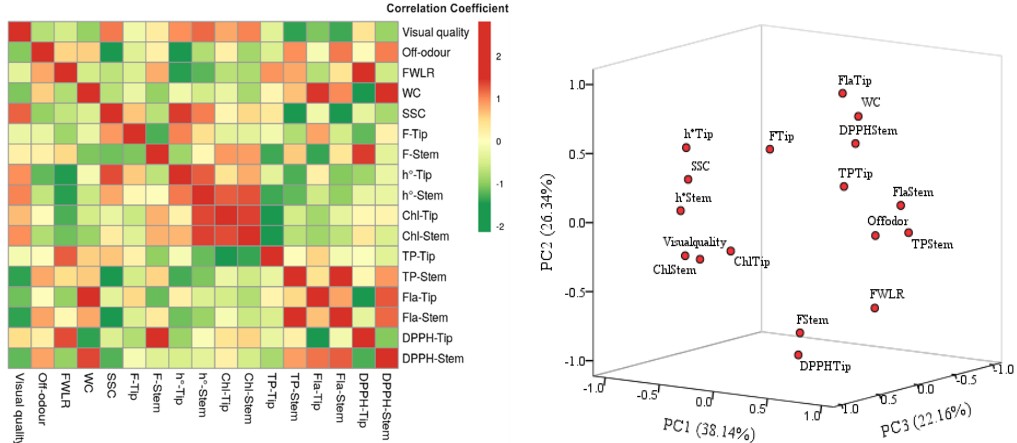

**Figure 5.** Pearson's correlation coefficient heatmap matrices and principal component analysis (PCA) among the responses of sensory, physiological quality attributes, and biochemical components in asparagus stored at 4 °C for 20 days of shelf life. FWL: fresh weight loss, WC: water content, SSC: soluble solids content, F-Tip: firmness in the tip, F-Stem: firmness in the stem, h°-Tip: Hue angle in the tip, h°-Stem: Hue angle in the stem, Chl-Tip: chlorophyll content in the tip, Chl-Tip: chlorophyll content in the stem, TP-tip: total phenolic content in the tip, TP-tip: total phenolic content in the stem, Fla-Tip: total flavonoid content in the tip, Fla-Tip: total flavonoid content in the stem, DPPH-Tip: DPPH radical scavenging activity in the tip, DPPH-Tip: DPPH radical scavenging activity in the stem.

The sensory quality of green asparagus was evaluated according to color, smell, decay, shrinkage, bract crack and others [50]. The association between sensory quality and the physiological situation was consistent with the sensory evaluation. These results are in agreement with Lu [51] who showed significant positive correlations between overall quality and firmness, SSC, TP, and negative correlations to the odor of 'Akihime' strawberry. Subsequently, principal component analysis (PCA) was performed and the results showed that the first three components (PC1, PC2, PC3) explained a total of 86.64% of the parameters of asparagus characteristic (Figure 5). The PC1, PC2, PC3 explained 38.14%, 26.34%, 22.16% of the variance, respectively. The visual quality, off-odor, TP-stem, Fla-Stem SSC, h° were clustered into the positive/negative of PC1, and the PC2 included FWL, DPPH, WC, Fla-Tip, F-Tip. Only F-Stem, TP-Tip, Chl-Tip variance belonged in PC3. The results confirmed the importance of visual quality, off-odor, firmness, color parameters, SSC and total phenolic content.

## 4. Conclusions

This study investigated the effects of elevated $CO_2$ pretreatment followed by modified atmosphere (Pre-MA) or micro-perforated (Pre-MP) packaging and continuous elevated $CO_2$ (Flow-$CO_2$) treatment

on the quality characteristics of green asparagus during storage at 4 °C. The development of microbial groups was inhibited by elevated $CO_2$ treatments, especially the Flow-$CO_2$ treatment, which showed the lowest counts of microorganisms. Although higher firmness was observed, elevated $CO_2$ pretreatments and Flow-$CO_2$ treatment resulted in better sensory quality, strong inhibition of microorganisms, high content of the nutritional and antioxidant activity, a higher content of SSC, phenolics, chlorophyll, and DPPH, and lower respiration rates. Furthermore, the results from the heatmap coefficient analysis and principal component analysis indicated that sensory quality, physiological attributes, and antioxidant activity responded differently depending on the specific treatments. The elevated $CO_2$ treatments caused changes in physiological attributes and content of biochemical components that determined the sensory qualities of green asparagus. Overall, Flow-$CO_2$ or Pre-MA treatments had a significant effect on the qualities of green asparagus and could be useful for green asparagus cold storage.

**Author Contributions:** Conceptualization and methodology, L.-X.W., I.-L.C., and H.-M.K.; Experiments performance and data curation, L.-X.W.; writing, L.-X.W.; writing—review and editing, H.-M.K. All authors read and agreed to the published version of the manuscript.

**Funding:** This study was supported by the IPET through Export Promotion Technology Development Program, funding from the Ministry of Agriculture, Food and Rural Affairs (Nos 117035-03).

**Acknowledgments:** The authors would like to thank lab members for their assistance.

**Conflicts of Interest:** The author declares no conflict of interest.

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
