# Peer review of "Correlations among Quality Characteristics of Green Asparagus Affected by the Application Methods of Elevated CO₂ Combined with MA Packaging"

_horticulturae, doi:10.3390/horticulturae6040103_

Round 1

Reviewer 1 Report

I've attached my edits for you to evaluate. Overly wordy at times with unclear meaning to sentences. Some sentences seem incomplete. Others need to be condensed. Use of "flowery" words (i.e. extremely, by nearly, such as) are filler words, are meaningless and detract from the paper.

Tighten up the writing and this will be a useful addition to the post-harvest literature on asparagus.

Reviewer 2 Report

This manuscript falls within the scope of this journal and addresses topics relevant to vegetable storage. The authors collected a good amount of data of suitable quality.

The manuscript is well structurated, the overall presentation is clear. Anyway before the acceptance of the manuscript some revisons need to be apported expeccially in the section of M&M.

In my personal opinion the title is no attractive, is too long and it is necessary to change it.

Line 40. Reference 2 is no pertinent for the paper, please chang eit.

Line 47: The emission of ethylene about strawberry is no correct because is no climateri fruirs so this adfirmation could be changed.

Line 49: please explicit HSPs when the authors use it for the first time.

M&M Avoid the use of approximately in teh text

Please describe the way to create the MAP. Is it an active MAP i suppose is no clear.

Results: please organize the sequence as reported in M&M.

Why the authors did’nt measeure the O2 also?
